# Investigation of Potency and Safety of Live-Attenuated Peste des Petits Ruminant Virus Vaccine in Goats by Detection of Cellular and Humoral Immune Response

**DOI:** 10.3390/v15061325

**Published:** 2023-06-05

**Authors:** Milovan Milovanović, Klaas Dietze, Ulrich Wernery, Bernd Hoffmann

**Affiliations:** 1Friedrich-Loeffler-Institut, Südufer 10, 17943 Greifswald-Insel Riems, Germany; milovan.milovanovic@fli.de (M.M.); klaas.dietze@fli.de (K.D.); 2Central Veterinary Research Laboratory, Dubai P.O. Box 597, United Arab Emirates; cvrl@cvrl.ae

**Keywords:** PPR, PPRV, goats, live-attenuated vaccine, IFN-γ, PPRV specific antibodies, cELISA, VNT, pen-side test, RT-qPCR

## Abstract

The peste des petits ruminant (PPR) virus is a transboundary virus found in small domestic ruminants that causes high morbidity and mortality in naive herds. PPR can be effectively controlled and eradicated by vaccinating small domestic ruminants with a live-attenuated peste des petits ruminant virus (PPRV) vaccine, which provides long-lasting immunity. We studied the potency and safety of a live-attenuated vaccine in goats by detecting their cellular and humoral immune responses. Six goats were subcutaneously vaccinated with a live-attenuated PPRV vaccine according to the manufacturer’s instructions, and two goats were kept in contact. Following vaccination, the goats were monitored daily, and we recorded their body temperature and clinical score. Heparinized blood and serum were collected for a serological analysis, and swab samples and EDTA blood were collected to detect the PPRV genome. The safety of the used PPRV vaccine was confirmed by the absence of PPR-related clinical signs, a negative pen-side test, a low virus genome load as detected with RT-qPCR on the vaccinated goats, and the lack horizontal transmission between the in-contact goats. The strong humoral and cellular immune responses detected in the vaccinated goats showed that the live-attenuated PPRV vaccine has a strong potency in goats. Therefore, live-attenuated vaccines against PPR can be used to control and eradicate PRR.

## 1. Introduction

The peste des petits ruminant virus (PPRV) is a single-strand nonsegmented negative-sense RNA virus that causes peste des petits ruminant (PPR) in sheep and goats, and it is currently found in Africa, Asia, and the Middle East with potential to spread to Europe [1,2,3]. The PPRV virus belongs to the genus Morbillivirus and encodes six structural proteins—(N), phosphoprotein (P), matrix (M), fusion (F), hemagglutinin–neuraminidase (HN), and large (L) protein— as well as two nonstructural proteins—C and V [4]. According to the results of a genetical analysis on N and F protein genes, the PPRV virus is genetically divided into four different lineages, whereas serologically, only one serotype exists [2,5]. The clinical signs of PPVR observed in infected small ruminants are mainly fever, mucopurulent oculonasal discharge, oral mucosal necrotic lesions, pneumonia, and gastroenteritis, which can range from mild to severe with lethal outcomes [6,7]. High morbidity and mortality rates are often observed in naive herds during epidemics, and lower mortality values are observed in endemic regions [8,9]. The occurrence of PPR has been noticed by the World Organization for Animal Health (WOAH). After Rinderpest, PPR is the second disease from the veterinary domain that has been targeted for eradication by 2030 as part of a global program coordinated by the WOAH and Food and Agriculture Organization (FAO) [10].

In controlling the spread of PPR to unaffected areas or populations, preventing animals from moving from affected areas is the main strategy, and rigorous quarantine and surveillance measures are used to enact this. To control and successfully eradicate PPR in endemic areas, small ruminants should be vaccinated according to previously described measures [10]. For the vaccination of sheep and goats, six different live-attenuated homologous vaccine strains called Nigeria 75/1 (lineage II), Sungri 96 (lineage IV), Arasur 87 (lineage IV), Coimbatore 97 (lineage IV), Titu (lineage IV), and 45G37/35-K PPR (lineage IV) are currently available [11,12]. Even though different live-attenuated vaccine strains exist, the strains that are most commonly used to vaccinate sheep and goats are Nigeria 75/1 and Sungri 96; they provide a measurable immune response for at least three years, do not cause notable side effects or the transmission of the vaccine, and effectively combat the virus [13,14]. A live-attenuated PPRV vaccine stimulates both innate and adaptive immune responses in vaccinated animals [15,16,17]. Innate immunity is the primary defense mechanism used to confront PPRV infections, and innate immunity is caused by pattern recognition receptors (PRRs), which recognize PPRV pathogen-associated molecular patterns (PAMPs). This reaction results in the production of several cytokines, and these develop an antiviral state in the host. The main group of cytokines responsible for this state are interferons (IFNs), which can be divided in two types: type I IFNs (IFN-α/β) are directly produced in response to the viral infection, whereas the type II IFN (IFN-γ) is secreted by activated T and natural killer cells [17]. The N protein of PPRV, which is a major structural protein, plays an important role in stimulating cell-mediated immunity, specifically the CD8+ and 4+ T-cell response, as well as the B cell response. In the early stage of infection, these activated T cells have the important role of recognizing the nonstructural C and V proteins of the virus before they are released from the infected cells via IFN-γ production or specific MHC-linked cytotoxicity [18]. High levels of N-specific antibodies are not neutralizing, and therefore, these antibodies have no purpose with regard to clearing the virus, virus particles, or infected cells, but they are indicators of an ongoing T-cell-mediated immune response. Two further structural proteins, H and F, are highly immunogenic and are also involved in stimulating the T-cell response, but more importantly, they are involved in stimulating the B-cell response and the formation of specific neutralizing antibodies. Compared to the N-specific antibodies, these H and F antibodies are crucial for the elimination of the virus, the virus particles, and the infected cells [14,15,18,19,20].

Different approaches are used to investigate and measure cytokine production or more specifically, INF production. The most commonly used tools are the enzyme-linked immunosorbent assay (ELISA) or enzyme-linked immunosorbent spot assay (ELISPOT), followed by the in vitro proliferation of peripheral blood mononuclear cells (PBMCs) [15,19]. After proliferation assays, transcriptome analysis via RNA sequencing is a very popular method that is used to investigate immune response modulation [16,21]. To detect PPRV-specific antibodies, different diagnostic assays are available, but the most commonly used assays are the virus neutralization test (VNT) and competitive ELISA (cELISA) [14,22,23]. Moreover, antigen detection has been remarkably enhanced by immunochromatographic lateral flow technology; it can be directly brought to the field, but for confirmation purposes, reverse transcription-polymerase chain reaction (RT-PCR) needs to be used [24,25].

Considering that PPR is still exotic in Europe, an outbreak would lead into severe direct losses due to the population being naïve, and affected countries will need to rely on the proper choice vaccines produced in Africa or Asia [11,26]. As the data regarding the potency and safety of commercially available PPRV vaccines in European small ruminant breeds not exists, a proper choice cannot be easily made. We investigated the potency of commercially available live-attenuated PPRV vaccines (Nigeria 75/1) in “Deutsche Edelziege” goats by subcutaneously (s/c) vaccinating six goats to determine their cellular (IFN-γ) and humoral (PPRV-specific antibodies) immune response. In addition to potency, we investigated the safety of the vaccine by keeping two naive goats in close contact and by detecting the PPRV genome with EDTA blood and different swab materials by using a molecular or easy-to-use rapid pen-side test. Our results provide important knowledge regarding the use of a live-attenuated PPRV vaccine in a European goat breed, and we emphasize the importance of the proper time point and diagnostic methods used to detect the cellular and humoral immune responses.

## 2. Materials and Methods

### 2.1. Experimental Design

#### 2.1.1. Experimental Animals and Vaccination

Eight male goats (breed: Deutsche Edelziege) that were approximately six months old that had originated from a farm in Germany were housed in a high-security containment facility at the Friedrich Loeffler Institute, Island of Riems. Six goats (Z/0, Z/1, Z/2, Z/3, Z/6, and Z/7) were subcutaneously (s/c) vaccinated with 1 mL of a live-attenuated PPRV vaccine that contained Nigeria 75/1 Pestvac (Jordan Bio Industries Center, Jovac, Jordanien) with a titer of 10^2.5^TCID_50_/mL according to the manufacturer’s instructions. The two remaining goats (Z/8 and Z/9) were kept together in the same facility as the in-contact goats to investigate the horizontal transmission of the vaccine virus from the vaccinated goats.

The goats were monitored daily during the whole length of the experiment, whereby we measured their body temperature and performed clinical examinations to discover clinical signs related to PPR by using a published clinical score system [7].

The reisolation and titration of the vaccine virus was performed on VERO dog SLAM cells (FLI cell culture collection number: RIE1280/57) in a T25 tissue culture flask and 96-well flat-bottomed tissue culture plate.

#### 2.1.2. Sampling

Blood, serum, nasal, conjunctival, and rectal swabs were collected before and after vaccination (−4, 3, 5, 7, 10, 12, 14, 17, 21, 24, and 28 dpv) and were used for the serological and virological analysis. 

All the blood samples were collected from the jugular vein of the goats by using an adapter system (Kabe Labortechnik GmbH, Germany). For serum collection, the tubes were centrifuged at 3000× *g* for 10 min and were stored at −80 °C until the analysis.

For the PBMC proliferation and IFN-γ production detection, heparinized blood was collected from the goats before and after their vaccination (−4, 3, 10, 17, and 24 dpv). After collecting the blood, the samples were kept at room temperature for 6 h before they were used for the PBMC stimulation.

### 2.2. Cell-Mediated Immune Response

In vitro PBMC stimulation was performed in a 96-well flat-bottomed tissue culture plate. After the heparinized blood was incubated at room temperature, each of the tubes was stirred by turning the tubes up and down few times, and 200 µL of the heparinized blood was placed in each well of one column (A1-H1). The stimulation was conducted in duplicate by using 20 µL of 1× phosphate-buffered saline (PBS) (negative control), 1 mg/mL of pokeweed mitogen (ID, Montpelier, France (positive control)), canine distemper virus (CDV) with an infectious titer of 10^2.5^TCID_50_/mL (unspecific antigen), and PPRV Nigeria 75/1 with an infectious titer of 10^3.25^TCID_50_/mL (specific antigen). After the addition of the stimulants, the plates were shaken for 3 min at 500 revolutions per minute (RPM) with a tilt shaker, and they were later placed into a humid chamber and incubated for approximately 20 h at 37 °C in 5% CO_2_. After incubation, the plate was centrifuged for 10 min at 500 g at room temperature, and 100 µL of plasma without erythrocytes was collected and stored at −80 °C until the analysis.

Interferon-γ (IFN-γ) production was measured by using a commercially available ID screen ruminant IFN-g ELISA kit (ID, Montpelier, France) according to the manufacturer’s instructions.

### 2.3. Humoral Immune Response

#### 2.3.1. Virus Neutralization Test (VNT)

To assess the titer of the PPRV-specific neutralizing antibodies, a VNT was conducted in 96-well flat-bottomed tissue-culture plates in a log2 dilution series in triplicate, whereby the positive, negative, and collected serum samples were tested against a constant titer (100 TCID_50_/mL) of the PPRV Nigeria 75/1 vaccine strain. Before titration, all the serum samples were incubated at 56 °C for 30 min for the complement inactivation. The serum samples were diluted (from 1:10 to 1:1280) in a serum-free cell-culture medium (FLI intern medium number: ZB5). The serum dilutions and a fixed amount of the PPRV Nigeria 75/1 vaccine virus were incubated for 2 h at 37 °C with constant gentle shaking by using a tilt shaker. After the neutralization step, a 100 µL suspension of the VERO dog SLAM cells (FLI cell culture collection number: RIE1280/57) in the ZB5 with 10% fetal calf serum (FCS) and 10 mg/mL of Zeozin (InvivoGen, Toulouse, France) was added in each well, and the plates were incubated at 37 °C and 5% CO_2_. Every time the test was performed, one control plate that contained the outcome of the back titration of the test virus and the titration of the positive and negative control serum was generated. After 4 days, the plates were inspected to see if the virus cytopathic effect (CPE) was visible, and a final reading was taken on day 7 when the results were recorded. The antibody titer was calculated by using the Spearman and Kaerber method [27]. The samples with a neutralizing titer greater than 10 were considered positive.

#### 2.3.2. Enzyme-Linked Immunosorbent Assay (ELISA)

A commercially available ID screen PPR competition ELISA kit (ID, Montpelier, France) was used to detect PPRV-specific antibodies from the serum samples according to the manufacturer’s instructions.

### 2.4. Peste des Petits Ruminant Virus Detection

#### 2.4.1. Pen-Side Test for PPRV Antigen Detection

Commercially available ID rapid peste des petits ruminant antigen dipstick field tests (ID, Montpellier, France) were used to detect PPRV antigens from the nasal and conjunctival swab samples according to the manufacturer’s instructions.

#### 2.4.2. Reverse Transcription Real-Time Polymerase Chain Reaction (RT-qPCR)

Viral nucleic acid was extracted from the sample material by using the NucleoMag VET kit (Macherey-Nagel, Düren, Germany) and the half-automated King Fisher platform (King-Fisher Flex magnetic particle processor, Thermo Fisher Scientific, Waltham, MA, USA). To amplify the conserved region of the nucleocapsid protein (Np), a specific primer probe mix [25] was employed with a FAM channel (concentration of the forward and reverse primer: 15 µM; probe: 5 µM). To control the extraction and amplification process, a heterologous EGFP-based control system [28] was implemented and codetected in all RT-qPCR runs by using the HEX channel (concentration of the forward primer, reverse primer, and probe: 5 µM).

The RT-qPCR reaction was performed by using commercially available AgPath-ID^TM^ One-Step RT-PCR Reagents of Applied Biosystems^TM^ (Waltham, MA, USA). Briefly, 2.5 µL of the template was added to the reaction mix, which contained 1.25 µL of RNase-free water, 6.25 µL of a 2× RT-PCR buffer, 0.5 µL of a 25× RT-PCR enzyme mix, 1.0 µL of a specific Np PPRV primer probe mix, and 1.0 µL of an EGFP mix primer probe mix. The RT-qPCR was run on a CFX 96 real-time PCR cycler (Bio-Rad, Hercules, CA, USA).

In every run, a generated PPRV standard series produced by the droplet PCR (QX200 Droplet Digital PCR System, Bio Rad, Hercules, CA, USA) was included to calculate the genome copy numbers.

## 3. Results

### 3.1. Vaccine Virus

The reisolation of the PPR vaccine virus on VERO dog SLAM cells was possible, and the cytopathogenic effect (CPE) of the virus in the form of syncytia was observed four days postinoculation. The successful reisolation of the virus was confirmed by the results of the PPRV-specific RT-qPCR. The original vaccine batch delivered a Ct value of 22.1, and the reisolated vaccine virus produced a Ct value of 14.6 in the PPRV real-time RT-qPCR. In addition, a direct titration of the PestVac vaccine was performed. According to the manufacturer, the titer expressed as the tissue culture infectious dose (TCID) of the vaccine virus should not be less than 10^2.5^TCID_50_/mL. For our used vaccine batch, a titer of 103,83 TCID50/mL was detected, which was in the line with our expectations.

### 3.2. Clinical Observation

The results of the daily investigations of the body temperature of the goats showed that the vaccinated goats (Z/3, Z/6, and Z/7) occasionally had a slightly high body temperature above 39.5 °C. When 18 dpv was used, all the vaccinated goats had a high body temperature: 39.7 °C (Z/7), 40.0 °C (Z/0, Z/3, and Z/6), 40.2 °C (Z/2), and 40.3 °C (Z/1). The in-contact goats had a normal body temperature through the duration of the study. Details on the daily body temperature of the vaccinated and in-contact goats are presented in Figure 1.

When looking for other clinical signs related to PPR, only clear seromucous nasal discharge was noticed in two of the vaccinated goats, Z/3 (12, 15, and 16 dpv) and Z/7 (from 9 to 15 dpv), and in one in-contact goat, Z/8 (11 dpv and from 13 to 16 dpv). Other clinical signs were not observed.

### 3.3. ID Screen Ruminant IFN-γ ELISA

The possibility of using heparinized blood for the PBMC stimulation was successfully demonstrated, as IFN-γ detection in samples stimulated with pokeweed mitogen (positive control) was possible on all stimulation days, and no IFN-γ production was detected in any of the samples stimulated with PBS (negative control). IFN-γ production as the response of the PBMC stimulated with naive PPRV (specific antigen) was detectable when 10 dpv was used in four vaccinated goats (Z-0, Z-1, Z-3, and Z-6). One of the two vaccinated goats (Z-7) that was negative for IFN-γ production after receiving 10 dpv reacted to the stimulation with PPRV by producing low amounts of IFN-γ (20) that were under the cutoff level. Neither the other vaccinated goat (Z-2) nor the in-contact goats had a reaction to the stimulation with PPRV, and IFN-γ production was not detectable at any time during the study. The test specificity was confirmed by the absence of IFN-γ production in all the goats during the CDV (unspecific antigen) stimulation. All the results are summarized and presented in Figure 2.

### 3.4. Humoral Immune Response

#### 3.4.1. Virus Neutralization Test (VNT)

Detecting specific PPRV-neutralizing antibodies in three of the vaccinated goats (Z-0, Z-1, and Z-6) was possible starting from 10 dpv. At 12 dpv, all the vaccinated goats seroconverted and had detectable PPRV-specific neutralizing antibodies. An increase in the antibody titer was observed in five of the vaccinated goats (Z/0, Z/1, Z/2, Z/3, and Z/6) until 21 dpv. One vaccinated goat (Z/7) showed a constant increase in the antibody titer from 12 to 28 dpv, and he had the highest detectable antibody titer out of all the vaccinated goats. Neither of the in-contact goats (Z-8 and Z-9) had detectable PPRV-specific neutralizing antibodies at any time during study. The detailed VNT results are presented in Figure 3.

#### 3.4.2. ID Screen PPR Competition ELISA (cELISA)

After using a commercially available cELISA to detect PPRV-specific antibodies, we found that after receiving 0, 3, and 5 dpv, two vaccinated goats (Z/3 and Z/7) had detectable values that were just slightly over the cutoff for questionable values. The detection of PPRV-specific antibodies in the vaccinated goats was possible in three goats (Z/0, Z/1, and Z/2) starting from 7 dpv, and we obtained clear positive results. At day 7 postvaccination, two goats (Z/3 and Z/6) had questionable values, and the remaining goat (Z/7) was negative. After receiving 10 dpv, all the vaccinated goats showed a strong antibody reaction, which lasted until the end of the trial. Neither of the in-contact goats (Z/8 and Z/9) demonstrated a detectable PPRV-specific antibody response at any time during the study. The results of the cELISA that was run on the vaccinated and in-contact goats are presented in Figure 4.

### 3.5. Virus Detection

#### 3.5.1. ID Rapid PPR Antigen Dipstick Field Test

All the nasal and conjunctival swab samples from the vaccinated and in-contact goats were found to be negative after the ID rapid PPR antigen dipstick field test was conducted. Detailed results of the ID rapid PPR antigen dipstick field test from the nasal and conjunctival swab samples can be found in Appendix A, respectively, in the Appendix A.

#### 3.5.2. Reverse Transcription Real-Time Polymerase Chain Reaction (RT-qPCR) Results of Swab and EDTA Blood Sample Material

After testing all the sample material, we detected a low genome load of PPRV in four vaccinated goats (Z/1, Z/2, Z/6, and Z/7) at different sampling time points. We could detect the PPRV genome in the vaccinated goats at 7 dpi by using EDTA blood for four goats (Z/1, Z/2, Z/6, and Z/7), using a nasal swab for three goats (Z/1, Z/6, and Z/7), and using a rectal swab for three goats (Z/1, Z/2, and Z/6). The time window for detecting the PPRV genome in the vaccinated goats was different when different sample materials were used, and for the nasal and rectal swab and EDTA blood, the dpv was up to 12 (Z/7), 14 (Z/6), and 17 dpv (Z/7), respectively. All the conjunctival swab samples from the vaccinated and in-contact goats were negative throughout the study period. Neither of the in-contact goats (Z/8 and Z/9) had a detectable PPRV genome load at any time during the study period. Detailed RT-qPCR results are presented in Figure 5.

## 4. Discussion

PPR outbreaks cause enormous direct and indirect costs worldwide. In addition to the high morbidity and mortality rates in goats and sheep, the easy spread of the disease, which is also driven by the human-behavior-induced mobility of small ruminants, is a particular challenge when it comes to controlling the disease [29]. A global eradication program, which is built on successful vaccination campaigns to a large extent, is highly dependent on up-to-date knowledge of the performance of the most-used vaccines. Thus, information on the host’s response to the vaccine, including potential vaccination virus shedding, is needed. As the diagnostic methods used to detect the virus and immune response are constantly being enhanced, reconfirming or adopting time windows for postvaccination monitoring tools continues to be important. The vaccination of small domestic ruminants at the population level continues to be a critical tool to ensure long-lasting immunity in vaccinated animals without remarkable side effects [23]. The vaccination of small domestic ruminants against PPR with live-attenuated vaccines is the most appropriate and widely used method for successfully controlling and eradicating the disease in endemic areas [11]. The results of research in which the authors investigated the immune response of goats and sheep following vaccination or infection have shown that the PPRV can stimulate both cellular and humoral immune responses by using different mechanisms [14,15,30]. We investigated the potency and safety of a commercially available live-attenuated PPR vaccine (Nigeria 75/1) in goats and determined a time window to detect their cellular immune response following vaccination; we performed this by measuring IFN-γ production and the humoral immune response of the goats by detecting PPRV-specific antibodies. Thus far, the results of all vaccination studies in which the authors have used live-attenuated PPRV vaccines of different lines to vaccinate sheep and goats have shown that the vaccines are safe at different doses and that no side effects such as fever or clinical signs associated with PPR occur [23]. For our study, all the goats were vaccinated according to the manufacturer’s instructions, and all the animals remained healthy over the study period. Occasionally, we observed slight elevations in the body temperature of three (Z/3, Z/6, and Z/7) vaccinated goats, but these elevations did not last longer than one day. Only after receiving 18 dpv did all the vaccinated goats have a high body temperature (39.7–40.3 °C) without any other clinical signs. This occasional rise in body temperature in a certain number of goats and in all the vaccinated goats after receiving 18 dpv cannot be attributed to their postvaccination reaction. The other clinical signs of a PPRV infection are conjunctival or nasal discharge, mucosal lesions, diarrhea, and respiratory symptoms [7], and two vaccinated goats and one in-contact goat had clean seromucous nasal discharge without congestion in their mucosal membranes. For one vaccinated goat (Z/3), discharge was intermittently observed; for another vaccinated goat (Z/7), discharge was observed for 7 days, and for an in-contact goat (Z/8), discharge was observed for 4 days. The progression of the nasal discharge from clean seromucous to mucopurulent with congestions, which has been described as a development pattern in PPRV-infected animals [7], was not observed. Similar clinical signs (nasal and ocular discharge) related to PPRV infection have been reported by other authors following the vaccination of small ruminants with a live-attenuated PPRV vaccine [14,22]. These clinical signs did not have a direct connection to the used vaccine virus and could not be assigned as a vaccination side effect. Most likely, the nasal discharge observed was due to PPRV-independent pathogens that caused local irritation in the nasal mucosa, but no systemic clinical signs such as fever or inappetence were evident. The influence of noninfectious causes such as low air humidity in animal facilities cannot be completely ruled out either.

To investigate the cell-mediated immune response of the goats, we measured IFN-γ production after PBMC stimulation in heparinized blood in 7-day intervals. Pokeweed mitogen (positive control) is described as a stimulant of choice for PBMC proliferation; more specifically, lymphocytes are stimulated during in vitro studies to measure IFN-γ production [31,32]. Our results are similar to the findings of other authors [31,32], as the all goats had high IFN-γ production throughout the study period in response to stimulation with pokeweed mitogen. The success of the stimulation was confirmed by the absence of IFN-γ production in all the goats stimulated with PBS (negative control). The test specificity was confirmed by the absence of IFN-γ production in all the goats stimulated with CDV (unspecific antigen) and the detection of IFN-γ production in only the vaccinated goats stimulated with naive PPRV (specific antigen), which was not related to the different titers of these two viruses. We found that different titers of the PPRV expressed as multiplicities of infection (MOI) will not reduce infectivity, and even if a low MOI is used, virus replication will be successful [33,34]. The results of the stimulation with PPRV showed that IFN-γ production was detectable in four out of the six vaccinated goats only after they received 10 dpv. Two other vaccinated goats were negative after receiving 10 dpv. After receiving 10 dpv, one vaccinated goat (Z/7) had low IFN-γ production, but the production was not high enough to be declared as positive compared with the other vaccinated goat (Z/2), who did not show any reactivity to the PPRV stimulation, as did both of the in-contact goats. Two vaccinated goats (Z/1 and Z/6) who had the highest IFN-γ production in response to the PPRV stimulation after receiving 10 dpv had low IFN-γ production after receiving 17 dpv, but these production levels were under the cu-off value determined by the manufacturer. The short time window and the limited number of vaccinated goats reacting to the PPRV stimulation by producing IFN-γ could be multifactorial; specifically, the dendritic cells might have not been sufficiently activated, the SLAM receptors had a limited expression, and the lineage of the virus used and goat breed might have influenced the results [15,22,33,35,36]. Dendritic cells are responsible for recognizing PPRV over toll-like receptor 2, which results in the proliferation and differentiation of T cells and the production of cytokines [15,36]. According to the findings of other authors, the detection of IFN-γ production under in vitro conditions following the stimulation of the isolated PBMC in vaccinated animals with PPRV is feasible after they receive 14 to 28 dpv [22]. Using phytohemagglutinin or concanavalin A to prestimulate the lymphocytes increases the SLAM receptor expression surface, which is reflected in the higher infectivity of PPRV, which can potentially lead to higher IFN-γ production [33,34,35]. Considering all the different factors, we suspect that the goats that did not react to the stimulation with PPRV had a lower activation of dendritic cells, which resulted in lower T-cell proliferation and differentiation. 

A strong humoral immune response was provoked by using a live-attenuated PPRV vaccine in the vaccinated goats, which is in the line with the findings of other authors [15,22]. According to the obtained serological results, a 50% seroconversion was observed in the goats after receiving 7 and 10 dpv when the cELISA or VNT was used, respectively. A 100% seroconversion was observed in the vaccinated goats after receiving 10 and 12 dpv when the ELISA or VNT was used, respectively, and seroconversion was not detected in the in-contact goats after both serological methods were used. The early detection of PPRV-specific antibodies in the vaccinated goats via cELISA and the VNT is in the line with the observations of other authors who used live-attenuated PPRV vaccine strains [14,15,22]. Comparing the results of the two different serological methods used to detect PPRV-specific antibodies, the ELISA test was more sensitive compared to the VNT. By more deeply investigating the development of PPRV-specific antibodies following vaccination with a live-attenuated PPRV vaccine, PPRV-specific antibodies against the N protein could be detected as early as after the goats received 7 dpv, whereas antibodies against the H protein could be detected after the goats received 14 dpv [14,22]. PPRV-specific antibodies against the N protein are not neutralizing compared to the antibodies against the H and F protein [18,19,20], which can explain the increased sensitivity of cELISA compared to the VNT. In one vaccinated goat (Z/7), a constant growth of PPRV-neutralizing antibodies was observed until the end of the experiment (28 dpv); this contrasted with the growth of the PPRV-neutralizing antibodies in the other vaccinated goats, which peaked at 21 dpv according to the VNT results. Following vaccination with a live-attenuated PPRV vaccine, the antibody production peak will be reached after 30 dpv is received and will be followed by a graduated decrease over a period of 3 years, with the defensiveness against the challenges associated with virulent PPRV being maintained [37]. As our study ended before the peak, the further rise in the antibody titer of this vaccinated goat (Z/7) could not be excluded. A constant increase in the PPRV-neutralizing antibodies in a period of 15 weeks following revaccination with live-attenuated PPRV Nigeria 75/1 was already observed in another study [15]. For our study, revaccination was not performed, but the prolonged detection of the PPRV genome load in the EDTA blood of this goat (Z/7) was observed with RT-qPCR. This finding could be associated with the prolonged antigen stimulation to produce PPRV-neutralizing antibodies.

The live-attenuated PPRV vaccine used in this study was shown to be safe in goats, as no transmission of the vaccine virus to the in-contact goats was confirmed. PPRV detection by using an ID rapid PPR antigen field test on the swab samples is mostly dependent on the availability of the antibody binding sites of the virus as well as the presence of a certain virus genome load (above 10^4^ genome copies) in the sample material [24,38]. Considering that these criteria are critical parameters for PPRV detection, the lack of antibody binding sites would explain the failure of antigen detection after use of the ID rapid PPR field test, as only low PPRV genome loads were detected with RT-qPCR in conjunction with the presence of PPRV-specific antibodies. Therefore, this rapid point-of-care test would not likely generate positive results in vaccinated animals in the field, which means that a positive test for an infection that is very likely to be a field virus infection requires control measures to be implemented.

We were able to detect the low PPRV genome load in four vaccinated goats (Z/1, Z/2, Z/6, and Z/7) by using the RT-qPCR method on nasal and rectal swabs and EDTA blood samples after the goats received 7 dpv. In one vaccinated goat (Z/7), we were able to detect the PPRV genome with RT-qPCR up until they received 17 dpv. Despite the detection of the PPRV genome in the swab sample material with RT-qPCR, the transmission of the used vaccine virus to the in-contact goats was not feasible, which confirms the safety of the used vaccine. For a prolonged amount of time, the PPRV genome was detected in the EDTA blood of a vaccinated goat (Z/7) who had an absence of fever and other PPR-related clinical signs; this was indicated by the constant rise in his antibody titer, which was found by conducting the VNT. These findings in vaccinated animals are new, as the authors of previous studies on PPR vaccination who used live-attenuated vaccines failed to detect the PPRV genome postvaccination [14,22]. The reason we were able to detect the PPRV genome in the sample material collected from the vaccinated goats is probably due to the higher titer of the vaccine used for vaccination compared with that used by other authors [14,22].

## 5. Conclusions

Our findings confirmed that the commercially available live-attenuated PPRV vaccine strain Nigeria 75/1 is potent and safe for the vaccination of “Deutsche Edelziege” goats. Negative pen-side test results, the detection of a low virus-genome load in sample materials from vaccinated goats according to RT-qPCR, the early immune response including seroconversion, and a lack of transmission to the in-contact goats confirmed that the excreted vaccine virus was not infectious and that horizontal transmission was not possible. Based on the data obtained, the long-term protection of the vaccination against PPRV field infections probably varies across individual animals. Since the immune response that has the greatest impact on vaccine efficacy is still unclear, a strong humoral immune response has been associated with the duration and extent of vaccine virus replication in the animal in general or in specific tissues. The strong cellular and humoral immune responses observed in a certain number of vaccinated animals can potentially be important to ensuring long-lasting protection. Altogether, we confirmed that the used commercially available live-attenuated PPR vaccine can be used to control and eradicate PPR in “Deutsche Edelziege” goats. Further investigations including different European small ruminant breeds and several commercially available live-attenuated PPRV vaccines are recommended, as this can contribute greatly to the ongoing PPR global eradication program.

## Figures and Tables

**Figure 1 viruses-15-01325-f001:**
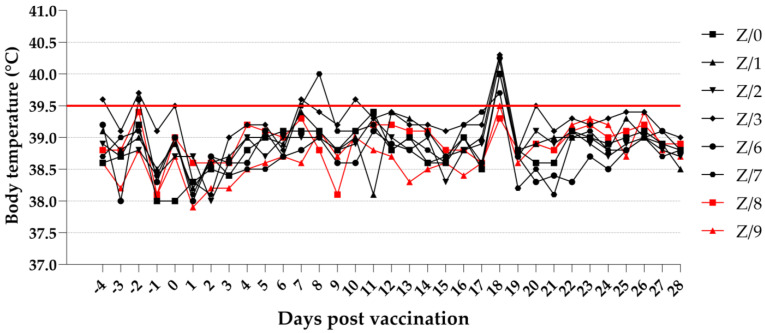
Daily body temperature of vaccinated (black symbols) and in-contact goats (red symbols). Red line cutoff (39.5 °C): normal body temperatures are under the cut off, while high body temperatures are above the cutoff.

**Figure 2 viruses-15-01325-f002:**
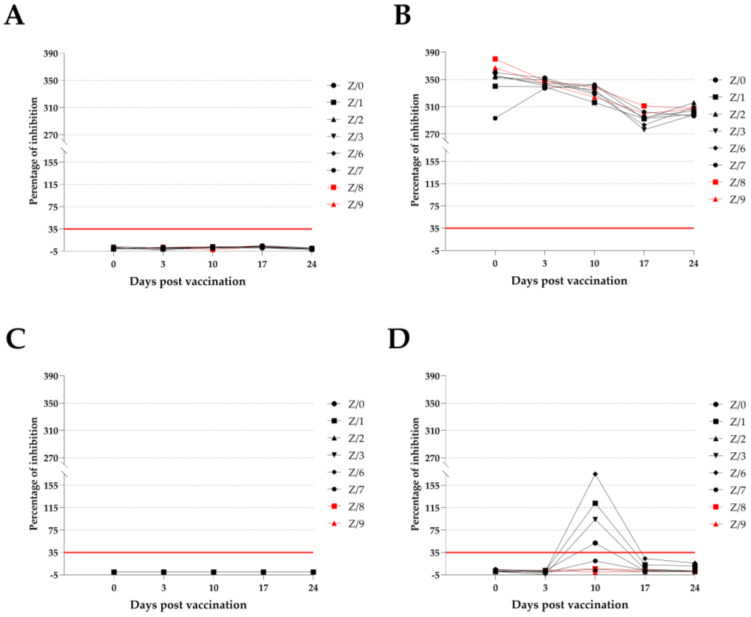
ID screen ruminant IFN-γ ELISA results as a percentage of inhibition of vaccinated (black symbols) and in-contact (red symbols) goats according to sampling day. Red line cutoff: values <35% are negative (under the red line), while those ≥35% are positive (above the red line). (**A**)—detected IFN-γ values in samples stimulated with PBS (negative control); (**B**)—with pokeweed mitogen (positive control); (**C**)—with CDV (unspecific antigen); (**D**)—with PPRV (specific antigen).

**Figure 3 viruses-15-01325-f003:**
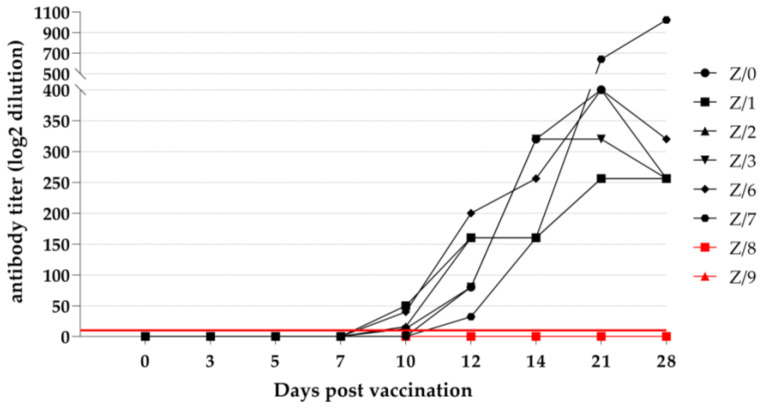
Virus neutralization test results of vaccinated (black symbols) and in-contact (red symbols) goats as a log2 dilution of analyzed serum samples. Red line cutoff: values of neutralizing titer ≥10 are positive (above the red line), while values of neutralizing titer <10 are negative (under the red line).

**Figure 4 viruses-15-01325-f004:**
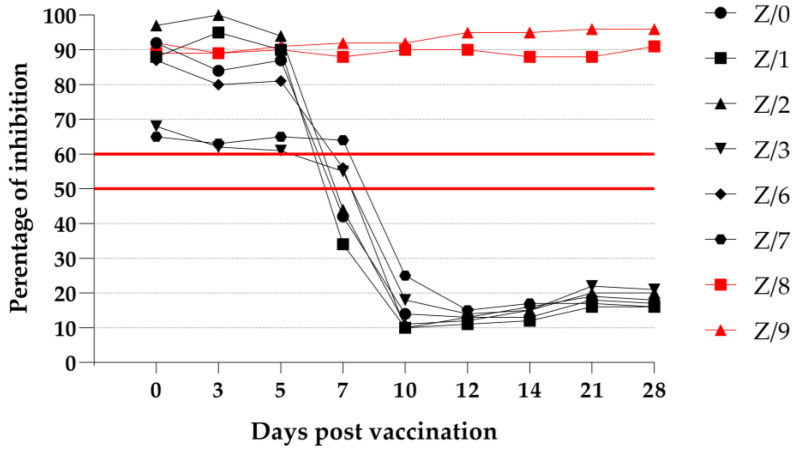
cELISA results as a percentage of inhibition of vaccinated (black symbols) and in-contact goats (red symbols). Values ≤50% are positive (under the lower red line), those >50% but ≤60% are questionable (between the red lines), and those ≥60% are negative (above the upper red line).

**Figure 5 viruses-15-01325-f005:**
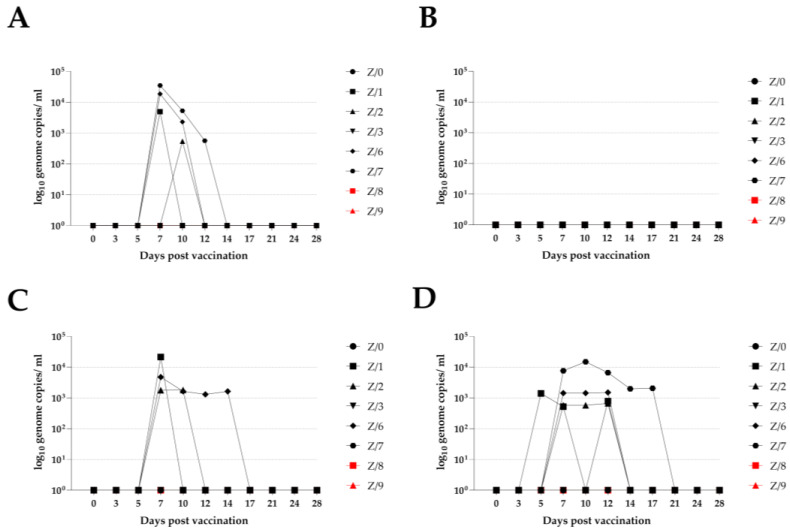
RT-qPCR results with PPRV genome load in vaccinated (black symbols) and in-contact goats (red symbols) according to sampling day. (**A**)—detected PPRV genome load in nasal swab; (**B**)—in conjunctival swab; (**C**)—in rectal swab; (**D**)—in EDTA blood.

## Data Availability

The data presented in this study are available upon request from the corresponding author.

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
