# Peer review of "Investigation of Potency and Safety of Live-Attenuated Peste des Petits Ruminant Virus Vaccine in Goats by Detection of Cellular and Humoral Immune Response"

_viruses, 2023, doi:10.3390/v15061325_

Round 1
Reviewer 1 Report
The topic selected for study - support for control of this economically and socially important disease of small ruminant animals - is a very important one.
The rationale for doing this study needs to be more clearly stated in the introduction. The authors identified determination of Nigeria 75/1 vaccine safety and potency as the objective of their study. Are previously published studies (references 14, 21) inadequate? What new information is needed and why?
The discussion needs to address these objectives in an organized way. Paragraphs should open with a statement of conclusion (This vaccine is safe because ...) followed by the interpretation of the findings that support that conclusion and a comparison with previously published findings. Limitations of the finding need to be mentioned. The vaccine elicits strong humoral and cellular responses ...
The details of their materials and methods are complete and clearly stated.
Some copy editing is required. For example:
Line 23. Live, not life.
Line 38. Severe
Several other places
Reviewer 2 Report
In this manuscript titled“Investigation of potency and safety of a live attenuated Peste des Petits Ruminant Virus vaccine in goats by detection of cellular and humoral immune response”, authors investigated potency of commercially available live attenuated PPRV vaccine (Nigeria 75/1) by subcutaneous (s/c) vaccination of six goats with detection of cellular (IFN-γ) and humoral (PPRV specific antibodies) immune response. Next, investigated vaccine safety by keeping two naïve goats in close contact and by detection of PPRV genome in different swab material using molecular or easy-to-use rapid pen-side test. The results confirmed that commercially available live attenuated PPRV vaccine strain Nigeria 75/1 is potent and safe for vaccination of Deutsche Edelzige goats. Providing important knowledge regarding the use of a live attenuated PPRV vaccine in goats emphasizing importance of proper time point and diagnostic method used for the detection of cellular as well as humoral immune response. There exists many minor problems in this manuscript, which need further revision and improvement. The specific amendments are as follows:
1. Some spaces need to be noticed, such as “Ulrich Wernery2” in line 5.
2. Please, check English grammar throughout the manuscript, such as “it's”in line 28 and “naïve” in. line 40.
3. In line 31, a genus should be italicized; In line 134, “In vitro” requires italics. Line 109 “Z/0, Z/1, Z/2, Z/3, Z/6 and Z/7”should be changed to “Z/0, Z/1, Z/2, Z/3, Z/6, and Z/7”ï¼›In line 123 “Blood, serum, nasal, conjunctiva and rectal swabs” should be changed to “Blood, serum, nasal, conjunctiva, and rectal swabs”.
4. Writing mistakes should be carefully checked, such as “102,5TCID50/mL” in line 140., “103,25TCID50/mL” in line 142, “10 mg/ml” in line 163,“minutes” should be changed “min”, in line 128. In “Figure 1” on line 219 and in “figure 2” on line 241. Is the negative “10” in line 261 different in font or bold?
5. Please make sure the full spelling of ELISA is “Enzyme linked immunosorbent test” or “Enzyme linked immunosorbent assay” In line 171. And in line 354, “seven”or “7”.
6. It is not necessary to use red and black for every figures, and the abscissa of Figure 1 can be widened a little, it is best to modify the horizontal coordinate values neatly.
7. Statistical analysis should be included in the materials and methods section.
8. The analysis method is simple and needs more innovation.
9. The quality of some Figures could be better. Please replace it with a higher resolution.
10. The results obtained in lines 281-282 lack corresponding data or charts as support.
11. Four small images appear in Figures 2 and 5, and the meaning represented by each small image should be explained.
12. There are many repeated abbreviations of proper noun in the manuscript, which should be deleted. For example, PBMC in lines 86 and 129.
13. It is mentioned in the manuscript:Progress of nasal discharge from clean seromucous to mucopurulent with congestions described as a development pattern in PPRV in- fected animals was not observed. What is the reason?
14. It is mentioned in the manuscript:Short window and limited number of vaccinated goats reacting to PPRV stimulation with production of IFN-γ observed in this study could be mul- tifactorial. Supplementary analysis can be considered how to control variable factors to the greatest extent to ensure the authenticity of the experiment.
15. There are too few references after 2018. It is suggested that the author add some cited references to give more support to theoretical data. For example, in sentences 301-304 lines.
16. Just listing the results is boring. The author can add some analysis or discussion about those results in the abstract to increase the study's novelty.
17. Compared with the results of other authors, the vaccine you used for vaccination has a higher titer, so you should explore the appropriate titer for vaccination.
18. Conclusions needs more in it, as it is more of an afterthought. The authors are suggested to highlight significant findings and include afterthoughts of this work.
good
Reviewer 3 Report
Please see the attached file for my comments to the authors.

Some typing errors to be corrected.
